

# Thermal regime, energy budget and lake evaporation at Paiku Co, a deep alpine lake in the central Himalayas

Yanbin Lei[1,2], Tandong Yao[1,2], Kun Yang[1,2,3], Zhu La[1], Yaoming Ma[1,2,4], Broxton W. Bird[5]

[1] Key Laboratory of Tibetan Environment Changes and Land Surface Processes, Institute of Tibetan Plateau Research, Chinese Academy of Sciences, Beijing 100101, China
[2] CAS Center for Excellence in Tibetan Plateau Earth System, Beijing, 100101, China
[3] Department of Earth System Science, Tsinghua University, Beijing 10084, China
[4] University of Chinese Academy of Sciences, Beijing, China
[5] Department of Earth Sciences, Indiana University-Purdue University Indianapolis (IUPUI), Indianapolis, IN 46202, USA.

*Correspondence to*: Yanbin Lei (leiyb@itpcas.ac.cn)

**Abstract.** Evaporation from hydrologically-closed lakes is one of the largest components of their lake water budget, however, its effects on seasonal lake level changes is less investigated due to lack of comprehensive observation of lake water budget. In this study, lake evaporation were determined through energy budget method at Paiku Co, a deep alpine lake in the central Himalayas, based on three years' in-situ observations of thermal structure and hydrometeorology (2015-2018). Results show that Paiku Co was thermally stratified between July and October and fully mixed between November and June. Between April and July when the lake gradually warmed, about 66.5% of the net radiation was consumed to heat the lake water. Between October and January when the lake cooled, heat released from lake water was about 3 times larger than the net radiation. Changes in lake heat storage largely determined the seasonal pattern of lake evaporation. There was about a 5 month lag between the maximum lake evaporation and maximum net radiation due to the large heat capacity of lake water. Lake evaporation was estimated to be 975±39 mm between May and December during the study period, with low values in spring and early summer, and high values in autumn and early winter. The seasonal pattern of lake evaporation at Paiku Co significantly affects lake level seasonality, that is, significant lake level decrease in post-monsoon season while slight in pre-monsoon. This study may have implications for the different amplitudes of seasonal lake level variations between deep and shallow lakes.

## 1 Introduction

Compared with numerous studies of inter-annual to decadal lake changes, seasonal lake level changes and the associated hydrological processes on the Tibetan Plateau (TP) are still less understood. Phan et al. (2012) showed that seasonal lake level variations in the southern TP are much larger than that in the northern and western TP. Song et al. (2014) further categorized eight regions with distinct patterns of lake level changes using cluster analysis. In-situ observations gave more details of seasonal lake level variations (Lei et al., 2017). One striking feature is the different amplitude of seasonal water level variations, that is, deep lakes usually exhibited considerably greater lake level variations than shallow lakes. For





example, Zhari Namco and Nam Co, two large and deep lakes on the central TP (Wang et al., 2009, 2010), exhibited

significant water level increase by 0.3~0.6 m during the summer monsoon season and a similar magnitude of lake level reduction by 0.3~0.5 m during post-monsoon season between 2010 and 2014. For the two nearby small and shallow lakes, Dawa Co and Bam Co, although there was a similar pattern of lake seasonality, the amplitude of seasonal lake level variations was considerably smaller than the two large and deep lakes (Lei et al., 2017).

Evaporation from hydrologically-closed lakes is one of the largest components of their lake water budget (Li et al., 2001;

Morrill, 2004; Xu et al., 2009; Yu et al., 2011). Unlike potential evaporation, the seasonal pattern of lake evaporation can be significantly affected by the heat capacity of lake water, especially for deep lakes. At Nam Co, for example, Haginoya et al. (2009) found that the sensible and latent heat fluxes were small during the spring and early summer, and increased considerably during the autumn and early winter due to the large heat capacity of lake water. Direct measurements of lake evaporation were usually conducted by using the eddy covariance system or energy budget method (e.g. Blanken et al., 2000;

Winter et al., 2003; Rouse et al., 2003, 2008; Rosenberry et al., 2007; Giannoiu and Antonopoulos, 2007; Zhang et al., 2014). On the TP, there are several studies regarding lake evaporation using the eddy covariance system, e.g Nogring Lake (Li et al., 2015), Qinghai Lake (Li et al., 2016), Nam Co (Wang et al., 2017; Lazhu et al., 2016), Siling Co (Guo et al., 2016). However, lake evaporation throughout the year is not well investigated on the TP because it is difficult to install and maintain measurement platform due to harsh natural conditions and the influence of lake ice during the late autumn and early

winter (Li et al., 2016). Furthermore, how lake evaporation affects seasonal lake level changes remains unclear due to lack of comprehensive observation of lake water budget.

To understand lake water budget and the associated lake level changes, we conducted a comprehensive in situ observation at Paiku Co (85°35.12′ E, 28°53.52′ N, 4590m a.s.l), a deep alpine lake in the central Himalayas (Wünnemann et al., 2015). The lake has a surface area of 280 km$^2$ and watershed area of 2376 km$^2$ (Nie et al., 2013; Dai et al., 2013). An investigation

of the lake's bathymetry and water level changes was previously conducted by Lei et al. (2018). They showed that the lake is a deep alpine lake with mean and maximum water depth of 41.1 m and 72.8 m, respectively. In this study, we first addressed the thermal regime and changes in heat storage at Paiku Co based on three years' water temperature profile data, then investigated hydro-meteorology and energy budget over the lake, and finally analyzed energy-budget derived lake evaporation throughout the year and its effect on lake level changes.

## 2 Material and Methods

### 2.1 Data acquisition

HOBO water temperature loggers (U22-001, Onset Corp., USA) were used to monitor changes in water temperature with an accuracy of ±0.2 oC. Two water temperature profiles were installed in Paiku Co's southern (0-42 m in depth) and northern (0-72 m in depth) basins (Fig. 1). In the southern basin, water temperature was monitored at the depths of 0.4 m, 5m, 10 m,

15 m, 20 m, 30 m and 40 m. In the northern basin, water temperature was monitored at the depths of 0.4 m, 10 m, 20 m, 40





m, 50 m, 60 m and 70 m. Since the lake level of Paiku Co fluctuates seasonally, the depth of water temperature loggers may also have changed in a range of 0.4-0.8 m. Water temperatures were recorded at an interval of 1 hour and daily-averaged values were used in this study. Three years' observational data from June 2015 to May 2018 from the southern basin was acquired, while only one year's data (June 2016 and May 2017) from the northern basin was acquired because the loggers

were lost in June 2018.

>>Fig. 1<<

To investigate local hydro-meteorology at Paiku Co, air temperature and specific humidity over the lake were monitored since June 2015 by using HOBO air temperature and humidity loggers (U12-012, Onset Corp., USA). The logger was installed in an outcrop ~2 m above the lake surface at the north part of the lake (Fig. 2). There was no data available between

February and May 2017 because the instrument battery was too low.

>>Fig. 2<<

Radiation, including downward shortwave radiation and longwave radiation to lake, was measured by Automatic Weather Station (AWS) at Qomolangma station for Atmospheric Environmental Observation and Research, Chinese Academy of Sciences (CAS). This station is located at the northern slope of Mount Everest, about 150 km east of Paiku Co (87 °1.22′E,

28 °25.23′N, 4276 m a.s.l). The 2 m air temperature, relative humidity, wind speed, radiation were recorded at an interval of 10 min. In this study, only downward shortwave radiation and longwave radiation are used. The climate conditions at Paiku Co and Qomolangma station were similar, including topography, altitude, cloud cover etc. Nonetheless, weekly averaged radiation was used to calculate lake evaporation in order to reduce the error caused by regional difference. The related information about hydro-meteorology observations at Paiku Co basin are listed in Table 1.

**2.2 Energy budget derived lake evaporation**

Lake evaporation was calculated using the energy budget (Bowen-ratio) method as described by Winter et al. (2003), Rosenberry et al. (2007), and Gianniou and Antonopouls (2007). The energy budget of a lake can be mathematically expressed as:

$$R = H + lE + S + G + A_v \qquad (1)$$

where R is the net radiation on the lake, H is the sensible heat flux from lake surface, lE is the latent heat utilized for evaporation, S is the change in lake water energy, G is the heat transfer between lake water and bottom sediment, and $A_V$ is the energy advected into lake water. The units used for the terms of Eq (1) are $W/m^2$. For large and deep lake, the components G and $A_V$ are small enough to be neglected, therefore we do not consider G and $A_V$ in our calculations.

The net radiation on the lake can be expressed as the following:

$$R = R_s - R_{sr} + R_a - R_{ar} - R_w \quad (2)$$

where $R_s$ is downward shortwave radiation, $R_{sr}$ is the reflection of solar radiation from lake surface, which is taken as 0.07 in this study (Gianniou and Antonopouls, 2007), $R_a$ is downward longwave radiation to lake, $R_{ar}$ is the reflected longwave





radiation from the lake surface, which is taken as 0.03, and $R_w$ is the longwave radiation from the lake surface. The units of the items in Eq (2) are W/m$^2$.

The longwave radiation from lake surface is approached by the equation:

$$R_a = \varepsilon_a \times \sigma \times (T_w + 273.15)^4 \quad (3)$$

where $R_a$ is the longwave radiation from lake surface, $\sigma$ is the Stefan-Boltzmann constant (=5.67×10$^{-8}$ Wm$^{-2}$K$^{-4}$), $\varepsilon_a$ is the atmospheric emissivity (0.97 for water surface) and $T_w$ is surface water temperature of the lake. Some studies show that the surface water temperature does not represent ''skin'' temperature, which is often derived from infrared thermometer (Rouse

et al., 2003, 2008). In this study, we use the water temperature at the depth of 0.4-0.8 m to represent the surface water temperature in lake evaporation calculation because surface water can be mixed quickly by wind in the afternoon.

The sensible heat flux is related to the evaporative heat flux through the Bowen ratio (Gianniou and Antonopouls, 2007):

$$\beta = \frac{H}{lE} = \gamma \times P \times \frac{T_s - T_a}{e_{sw} - e_d} \qquad (4)$$

where $T_s$ is the surface water temperature, $T_a$ is air temperature at 2m high above the water surface, $e_{sw}$ and $e_d$ are the

saturated vapor pressure at the temperature of the water surface and the air vapor pressure above the water surface (kPa), respectively, P is air pressure (kPa), and $\gamma$ is the psychrometric constant, 6.5×10$^{-4}$/°C. In this study, air temperature, air pressure and specific humidity were monitored at the lake's shore. Saturated vapor pressure at the lake surface was calculated according to surface water temperature in the southern center of the lake. To match the radiation, all the input data were weekly averaged before lake evaporation was calculated.

Changes in lake heat storage (S) were calculated according to the detailed lake bathymetry and water temperature profile:

$$S = \sum_{i=0}^{72.8} \Delta V_i \times \Delta T_i \quad (5)$$

Where $\Delta V_i$ is the lake volume at certain depth, and $\Delta T_i$ is water temperature change at the same depth. The lake volume was calculated at an interval of 5 m according to the isobath of Paiku Co (Lei et al., 2018). $\Delta T$ was also calculated at an interval of 5 m as the average value of the top and bottom water temperature. Changes in heat storage for the bottom water (>40 m)

in 2015/2016 and 2016/2017 were calculated according to the data in 2016/2017 since there is no data in the other two years.

## 3 Results

### 3.1 Thermal structure of lake water

The three years' water temperature profile data shows that Paiku Co was stratified between July and October, and fully mixed between November and June in each year of the study period (Fig. 3). The lake water was heated by solar radiation

between April and June when water temperature increased rapidly from 1 °C to 7 °C. During this period, slight temperature differences between the surface and bottom water appeared, but this difference was less than 1 °C and can not prevent lake water from mixing due to strong wind turbulence and convective current (Wetzel, 2001). The temperature gradient increased

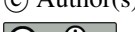



dramatically in late June with clear stratification occurring in July, which corresponded to a significant reduction in wind

speed (data not shown). Strong lake surface heating and the reduction in wind speed together contributed to the development

of thermal stratification in the water column. This is because rapid lake surface warming serves to weaken convection

currents while reduction in wind speed effectively reduces water column turbulence (Wetzel, 2001). During the summer

stratification period, the surface water warmed up rapidly from 7 to ~13 $^o$C between July and August, while the bottom water

warmed slowly. For example, the water temperature at 40 m water depth only increased by 1$^o$C between July and August.

There was almost no change in water temperature at 70 m water depth during the same period. As a result, the thermocline

formed between 15 m and 30 m water depth, with the largest temperature difference (5~6 $^o$C) occurring in August.

Due to the decrease in solar radiation, the lake surface began cooling in September, however, the bottom water continued to

slowly warm (Fig. 3). As a result, the water temperature gradient decreased, which caused the lake stratification to break

down in late October of each year. Notably, the timing of the stratification breaking down corresponded well to significantly

increased wind speed during this time (data not shown). The cooling of surface water can lower the temperature gradient and

hence thermal stability of the water column, while the dramatic increase in wind speed can strengthen the mixing of lake

water, further destabilizing the water column. Unlike the rapid appearance of lake stratification in late June, the breakdown

of stratification occurred more gradually, with the mixing layer deepening gradually throughout October (Fig. 4). The

mixing layer reached to 40 m water depth on 13th Oct, 2016, while it reached to 70 m water depth about half a month later

on 30th Oct. Following the complete breakdown of the water column's stratification, the bottom water experienced rapid

warming in several days due to its mixture with the warmer water from the upper layer. For example, the water temperature

at 70 m water depth remained stable at ~6.9 $^o$C from July to October, but increased abruptly from 6.9 $^o$C to 8.6 $^o$C from 25th

October to 30th October.

>>Fig. 3<<

>>Fig. 4<<

Water temperature of the whole lake decreased gradually from 8.6 to 1 $^o$C from November to January and remained stable at

~1 $^o$C in February. The identical lake water temperature profiles at the two monitoring sites indicated that Paiku Co's water

column was fully mixed between November and May (Fig. 3, Fig. 4). The thermal structure at Paiku Co is similar to

Bangong Co (Wang et al., 2014) and Nam Co (Lazhu et al., 2016), but different from Dagze Co (Wang et al., 2014). The

identical lake water temperature profile at Paiku Co indicates that changes in lake heat storage are not only affected by

surface water, but also bottom water. For deep lakes like Paiku Co, changes in heat capacity can be significantly

underestimated if only the surface water is considered.

### 3.2 Spatial difference of lake water temperature

Spatial difference of lake water temperature was investigated using in-situ observation of lake water temperature at different

sites. First, we compared water temperature difference between Paiku Co's southern and northern basins (Fig.5). Since the

northern basin is much deeper than the southern basin, lake water at the northern basin warms more slowly than that at the





southern basin during the spring and early summer, and cools more slowly during the autumn and early winter. Taking an example of the lake water temperature in 2016/2017, the surface water temperature in the southern basin was about 0.85 $^{o}$C higher on average than that of the northern basin between April and September. The water temperature became spatially uniform in late October when the water column was fully mixed. During November and December when the water

temperature decreased, the surface water temperature in the southern basin was about 0.45 $^{o}$C lower on average than that in the northern basin. Water temperature became spatially uniform in both basins again between January and March. This spatial difference was also found to at the depth of 10 m (Fig.5).

A contrasting pattern of water temperature changes can be found in the hypolimnion. Between mid-August and mid-September, water temperature at 20 m depth was about 0.81 $^{o}$C lower in the southern basin than that in the northern, which

contrasts with that of the surface water characteristics (Fig. 5c). Similar conditions occurred at 40 m depth, when water temperature was 0.75 $^{o}$C lower in the southern basin relative to the northern basin between mid-September and mid-October (Fig. 5d). This contrasting pattern of water temperature between surface and bottom water may be mainly attributed to water circulation along the south-north transection. Water convection became deeper due to the decrease in water temperature gradient in autumn.

We further compared the water temperature measured in the lake center with those measured along its shoreline. Water temperature along the northern and eastern shoreline of Paiku Co was recorded by HOBO water level loggers (Fig. 1). The results show that the water temperature along lake's shore was very sensitive to air temperature and fluctuated with much larger amplitude than that in the center of the lake, although both of them exhibited similar seasonal fluctuations (Fig. 5E, 5F). For example, shoreline water warmed more quickly to higher temperature during the spring and summer as compared

with that temperature in the lake's center, but conversely the shoreline water cooled more quickly in the autumn. The spatial difference of water temperature indicates that large errors can result if water temperature data collected at shoreline are used to calculate lake evaporation.

\>\>Fig. 5\<\<

### 3.3 Lake hydrometeorology

The annual mean air temperature over Paiku Co was 4.7 $^{o}$C in 2016 with the highest air temperature in July (11.2 $^{o}$C) and the lowest in January (-2.4 $^{o}$C). Due to the large heat capacity of lake water, Paiku Co can serve as an 'air conditioner' and moderate seasonal temperature variability of the overlying atmosphere. In the spring and early summer (April and June), lake water can significantly cool the overlying atmosphere, with air temperature at Paiku Co 1.5 $^{o}$C lower on average than that at Qomolangma station (Fig. 6). In the autumn and early winter, lake water can dramatically warm the overlying

atmosphere (October and December), with air temperature at Paiku Co 2.1 $^{o}$C higher on average when compared with those at Qomolangma station. There was a ~1.5 months lag between lake surface water temperature and air temperature, with the highest air temperature occurring in late August and the lowest in February (Fig. 6). The temperature difference between the lake surface and the overlying atmosphere exhibited a linear increasing trend from June to November, and a linear





decreasing trend from January to June. Positive temperature difference mainly occurred during the autumn and winter, while
negative difference occurred during the spring and early summer.

Atmospheric water vapor content at Paiku Co exhibited high value from June to September (Fig. 6), which was consistent
with the occurrence of Indian summer monsoon precipitation. During the non-monsoon season (October to May), the
atmospheric water vapor content was generally low. The water vapor pressure difference between the lake surface and the
overlying atmosphere exhibited a linear increasing trend from June to September and then a linear decreasing trend from
October to February. High water vapor difference occurred between September and December (0.76 kPa), while low
difference was observed between March and June (0.39 kPa).

>>Fig. 6<<

Radiation, including downward shortwave radiation, downward longwave radiation to lake and upward longwave radiation
from the lake body, is the main driver of lake energy balance. Downward shortwave radiation at Paiku Co had an annual
average of 251.8 W/m$^2$ (Fig. 7), which are slightly higher than TP average due to the lower latitude (Yang et al., 2009).
Downward longwave radiation to lake had an average of 235.8 W/m$^2$. The total radiation income was always higher than
upward longwave radiation from the lake body, which had an annual average of 336.8 W/m$^2$. The net radiation over Paiku
Co varied seasonally between 19.0 and 212.1 W/m$^2$, with an average value of 125.8 W/m$^2$. Relatively high net radiation
occurred from April to August (200.4 W/m$^2$), with the highest value in June (212.1 W/m$^2$). Relatively low net radiation
occurred from October to February (52.2 W/m$^2$), with the lowest value in December (19.7 W/m$^2$).

>>Fig. 7<<

### 3.4 Impact of lake heat storage on total heat flux

Changes in lake heat storage were quantified according to in-situ observation of water temperature profile and detailed lake
bathymetry, which makes it possible to evaluate the impact of lake heat storage on the total heat flux (Fig. 8). Between April
and July when Paiku Co warmed gradually, the lake water absorbed energy at an average rate of 128.6 W/m$^2$, accounting for
66.5% of net radiation during the same period. The lake heat storage increased most rapidly in June, with an average rate of
191.6 W/m$^2$, accounting for 91.6% of net radiation. The lake heat storage reached its peak in late August, when the surface
water temperature was highest. Between October and January when Paiku Co cooled, the lake water released energy at an
average rate of 137.5 W/m$^2$, which was more than 3 times larger than net radiation. The lake released energy most rapidly in
November, at an average rate of 193.6 W/m$^2$, which was about 5 times larger than net radiation.

The total heat flux over the lake surface, including the latent and sensible heat fluxes, was determined as the difference
between net radiation and changes in lake heat storage. Due to the large heat storage of lake water, the seasonal pattern of
total heat flux at Paiku Co was significantly affected by changes in heat storage, rather than the net radiation. There was a ~5
month lag between the maximum total heat fluxes and maximum net radiation. The lowest total heat flux occurred in June
and the highest in November. Fig. 8 shows a comparison of weekly averaged net radiation, changes in lake heat storage, and
heat fluxes over the lake surface. Although net radiation was high in spring and summer, a large portion of energy was





consumed to heat lake water, which resulted in low heat fluxes. In the autumn and early winter, although net radiation was relatively low, a large amount of heat stored in the lake was released into the overlying atmosphere, which resulted in high heat fluxes.

>>Fig. 8<<

### 3.5 Sensible and latent heat fluxes

The Bowen ratio determines the distribution of sensible and latent heat fluxes. At Paiku Co, the Bowen ratio varied in a range of -0.26~+0.37, with an annual average value of +0.08 (Fig. 9, Table 2). Negative value occurred between April and July, with an average value of -0.12, indicating the lake water absorbed energy from the overlying atmosphere. Positive

value occurred between August and January, with an average value of 0.20, indicating the lake water released energy to the overlying atmosphere.

Latent heat flux, with an average value of 112.3 W/m$^2$, was the main component of heat flux, accounting for 90% of the total heat flux between May and December. The latent heat flux varied similarly with total heat flux. During May and June, the latent heat was low with an average of 38.7 W/m$^2$. Between October and December, the latent heat was high, with an

average of 153.3 W/m$^2$.

Sensible heat flux, with an annual average value of 13.3 W/m$^2$, only accounted for 10% of the total heat flux. There was a high correlation between sensible heat and the water temperature difference between surface water and the overlying atmosphere (r=0.93). The sensible heat was negative between April and July with an average value of -5.6 W/m$^2$ (Fig. 9b), which was mainly due to the negative temperature difference between surface lake water and the overlying atmosphere. The

sensible heat flux became positive at the end of July, and between August and December, the sensible heat was high, with an average value of 23.0 W/m$^2$. The highest sensible heat flux occurred in October with an average value of 29.2 W/m$^2$ due to the large temperature difference between surface lake water and the overlying atmosphere.

>>Fig. 9<<

### 3.6 Lake evaporation

As shown in Fig. 10, lake evaporation was generally low in May and June with an average value of 1.7 mm/day. In July and August, lake evaporation increased rapidly from 2.9 to 4.1 mm/day. High lake evaporation occurred between September and December, with an average value of 5.4 mm/day. The total lake evaporation between May and December was estimated to be 975 mm during the study period.

>>Fig. 10<<

Lake evaporation at Paiku Co lagged net radiation by about five months and exhibited a similar seasonal pattern with changes in lake heat storage, indicating that the large heat capacity of lake water played an important role in the seasonal pattern of lake evaporation. Regression analysis shows that lake evaporation at Paiku Co positively correlated with changes in lake storage (r$^2$=0.63, P<0.001). However, lake evaporation negatively correlated with net radiation (r$^2$=0.22, P<0.001),





indicating the large heat storage significantly changed the seasonal pattern of lake evaporation. Lake evaporation also
exhibited similar patterns with the water vapor pressure difference between surface water and the overlying atmosphere
($r^2$=0.33).

## 4 Discussion

### 4.1 Uncertainty of lake evaporation estimation

There are several factors that can cause uncertainty in estimating lake evaporation. The first one is the determination of solar
radiation and atmospheric long wave radiation at Paiku Co. In this study, solar radiation and atmospheric long wave
radiation at Qomolangma station, which is about 150 km away from Paiku Co, were used to represent values at Paiku Co. To
evaluate the spatial difference, we made a comparison of solar radiation at Paiku Co and Qomolangma Station by using
Hamawari-8 satellite data (data not shown). The results show that daily solar radiation at the two sites exhibited very similar
seasonal fluctuations ($R^2$=0.55, P<0.001), with mean deviation of 1.3 W/m$^2$. Assuming approximately 70% of the net
radiation was consumed by lake evaporation (Lazhu et al., 2016), the uncertainty of lake evaporation due to error in solar
radiation was ~12 mm per year ($\Delta E_1$).

Despite the slight difference in solar radiation between Paiku Co and Qomolangma station, it is still difficult to accurately
evaluate solar radiation at Paiku Co due to the complex terrain surrounding the lake. The actual solar radiation at Paiku Co
can be considerably overestimated due to the blocking effect of the surrounding mountains around the lake. Therefore, lake
evaporation in this study may be overestimated to some extent. Lake level decrease between October and mid-January can
be used to validate lake evaporation because both surface runoff and precipitation are already very low in this high mountain
region. We found that lake evaporation (5.4 mm/day) is considerably higher between October and January than the rate of
lake level decrease (3.8 mm/day). While this discrepancy may be due to the overestimation of solar radiation and lake
evaporation, it is also possible that other factors, including precipitation, limited surface runoff (Tab. 3), and groundwater
recharge, contributed to the lake water balance, thereby slightly offsetting some of lake evaporation.

The second factor affecting the estimation of lake evaporation is lake water temperature. The water temperature profile was
monitored in Paiku Co's southern and northern basins. In this study, lake water temperature data in the southern basin was
used to determine the seasonal changes in lake evaporation. As we have shown in Fig. 5, although there were similar
seasonal fluctuations, water temperature exhibited considerably spatial differences between the lake's southern and northern
basins. To estimate the uncertainty of lake evaporation, we further calculated lake evaporation at Paiku Co by using the same
lake hydro-meteorology data, but water temperature profile in the northern basin in 2016. The difference of lake evaporation
between the two sites can be roughly taken as one of the uncertainties of lake evaporation at Paiku Co.

Lake evaporation using water temperature from the northern basin was estimated to be 911 mm from June to December 2016,
which was 20 mm larger compared with that estimated for the southern basin (891 mm). The largest difference in lake
evaporation between these two sites was in June and November. The accumulated lake evaporation from the northern basin





was 51 mm higher in June than evaporation from the southern basin, but 41 mm lower in November. Different heat capacity in the southern and northern basins determined the energy distribution that can be used to evaporate lake water. Assuming similar error of lake evaporation between May and June, the uncertainty of lake evaporation caused by water temperature difference was estimated to be about 36 mm ($\Delta E_2$). Thus, the total uncertainty of lake evaporation was estimated to be 39

mm (= $\sqrt{\Delta E_1{}^2 + \Delta E_2{}^2}$), accounting for 4.0% of total evaporation.

## 4.2 Comparison of lake evaporation with other lakes on the TP

To further explore the impact of lake heat capacity on the seasonal pattern of lake evaporation, we compared lake evaporation at Paiku Co with other lakes on the TP. We only selected lakes with direct measurements of lake evaporation, including the eddy covariance system or energy budget method. At Ngoring Lake (area, 610 km$^2$; mean depth, 17 m) on the

eastern TP, Li Z. et al. (2015) investigated the lake's energy budget and evaporation in 2011-2012 using the eddy covariance system, and found that the latent heat at Nogring Lake was lowest in June, peaked in August and then decreased gradually from September to November. At Qinghai Lake (area, 4430 km$^2$; mean depth, 19 m) on the northeast TP, Li X. et al. (2016) conducted studies concerning the lake's energy budget and evaporation in 2013-2015 using the eddy covariance system, and found that there was a 2–3 month delay between the maximum net radiation and maximum latent and sensible heat fluxes.

Compared with the two larger but shallower lakes, there was longer time lag between the total heat flux and net radiation at Paiku Co. As we have shown, Paiku Co has the mean water depth of ~41 m and the water column is fully mixed between November and June. This means that the lake can store more energy in spring and early summer than shallow lakes, and can release more energy to the overlying atmosphere in the autumn and early winter.

At Nam Co, a large and deep lake on the central TP, there have been several studies regarding lake evaporation (Haginoya et

al., 2009; Ma et al., 2016; Wang et al., 2016). Haginoya et al. (2009) found that lake evaporation at Nam Co was lowest in May and highest in October. The Bowen ratio-derived lake evaporation was estimated to be 916 mm in 2013 (Lazhu et al. 2016). Comparison with Paiku Co shows that both lakes exhibited similar seasonal pattern of lake evaporation, although lake evaporation at Paiku Co was slightly larger than that at Nam Co due to its higher solar radiation. In fact, although the maximum depth at Nam Co is greater than at Paiku Co, the average water depth of the two lakes is similar(Wang et al., 2009;

Lei et al., 2018), which resulted in the seasonal pattern of evaporation at the two lakes being comparable. At Siling Co, another large and deep lake on the central TP, monthly lake evaporation was found to vary within a range of 2.4-3.3 mm/day between May and September, 2014, with a total amount of 417.0 mm (Guo et al., 2016). Although the accumulative evaporation between Paiku Co and Siling Co was similar between May and September, lake evaporation at the two lakes between October and December can not be further compared because the energy flux at the lake was not measured at Siling

Co.



## 4.3 Implications for the amplitude of seasonal lake level variations at different lakes

The quantification of lake evaporation is important for understanding lake water budget and associated lake level changes. Compared with the eddy covariance system that can only work until October/November when the lake surface begins to freeze up (Li et al., 2015; Wang et al., 2017; Guo et al., 2016), our results give a full description of lake evaporation during
the entire ice-free period. More importantly, our results indicate that for deep lakes on the TP, evaporation during the post monsoon season can be much higher than that during the pre-monsoon and monsoon seasons due to the release of large amount of stored heat (Haginoya et al., 2009), despite both air temperature and net radiation are much lower. In this sense, lake evaporation during the cold season (October to December) is of great importance to lake water budget and can significantly affect the amplitude of lake level changes.

As shown by Lei et al (2018), lake level at Paiku Co decreased considerably by 3.8 mm per day on average between October and December, which is in contrast to the slight decrease of 1.3 mm per day between mid-April and May. Lake evaporation can explain the different rates of lake level changes between spring and autumn. In-situ observations of runoff at the three main rivers indicate that the surface runoff had weak impact on lake level changes during pre and post monsoon seasons (Tab. 3). High lake evaporation rates between October and December (5.4 mm/day) corresponded well to the rapid lake level
decrease (3.8 mm/day), while low lake evaporation in May (1.7 mm/day) corresponded to the reduced lake level decreases (1.3 mm/day). This suggests that lake evaporation largely determined the amplitude of lake level changes in dry seasons.

In a larger sense, these results may be applicable to other lakes on the TP. Lei et al (2017) investigated the lake level seasonality across the TP and found that there were different amplitudes of lake level fluctuations even in similar climate regimes. For example, lake level at Nam Co and Zhari Namco, two large and deep lakes on the central TP, decreased
considerably by 0.3-0.5 m in cold season (October to December), while lake level at two nearby small lakes, Bam Co and Dawa Co, decreased slightly by of 0.1-0.2 m during the same period. For deep lakes (e.g. Paiku Co, Nam Co and Zhari Namco), the latent heat flux (lake evaporation) over lake surface may lag the solar radiation by several months due to the large heat capacity of lake water. For this kind of lake, the lake level drop is most dramatic in the autumn and early winter when lake evaporation is high. For shallow lakes, the latent heat flux closely follows solar radiation, with high lake
evaporation during pre-monsoon and monsoon seasons, and low lake evaporation during post monsoon season (Morrill et al., 2004). Meanwhile, the shallow lakes freeze up 1-2 months earlier than deep lakes. When the lake surface is covered by ice, lake evaporation is effectively prohibited. Consequently, lake level decreased at a much slower rate in shallow lakes compared with deep lakes. This phenomenon can also be seen in some thermokarst lakes on the northern TP (Luo et al., 2015; Pan et al., 2017).

## 5 Conclusion

As an important component of lake water budget, lake evaporation was investigated through energy budget method at Paiku Co, a deep alpine lake in the central Himalayas, based on three years' in-situ observations of lake water temperature profile



and hydrometeorology between June 2015 and May 2018. The results show that Paiku Co was stratified from July to October, and totally mixed between November and June. The surface water were warmest during late August, while the

bottom water temperature increased more slowly, reaching to its highest suddenly at the end of October when the water column finally mixed. The lake water temperature was coolest in February.

As a deep alpine lake, changes in lake heat storage largely determined the seasonal pattern of lake evaporation. The lake absorbed 66.5% of net radiation to heat the lake water in the spring and early summer and released this energy to the overlying atmosphere in the autumn and early winter. Between October and January heat released from lake water was about

3 times larger than the net radiation. As a result, there was about a 5 month lag between the maximum total heat flux over the lake surface and the maximum net radiation due to the large heat capacity of lake water. Lake evaporation was estimated to be 975±39 mm between May and December during the study period, with low values between May and June (1.7 mm/day), and high values between September and December (5.4 mm/day).

This study may have implication for the different amplitude of lake level seasonality between shallow and deep lakes. For

deep lakes like Paiku Co, high lake evaporation occurs in the post monsoon season (October to December), which leads to the rapid decrease in lake water level. While low lake evaporation occurs in pre-monsoon season, which leads to slight lake level decrease. For shallow lakes, the seasonal pattern of lake evaporation varies similarly with net solar radiation, which results in only slight lake decrease in post-monsoon season and less amplitude of lake level seasonality.

**Data availability**

All original data presented in this paper are publicly available via National Tibetan Plateau Data Center (http://data.tpdc.ac.cn/en/).

**Author contribution**

LeiY.B. and Yao T.D. conceived and designed the experiments; Lei.Y., YaoT.D., Yang K., Lazhu, and Ma Y.M. analyzed the data; LeiY.B. performed the fieldwork and wrote the paper; Bird B.W. polished the language.

**Competing interests**

The authors declare that they have no conflict of interest.





## Acknowledgement

This research has been supported by the Strategic Priority Research Program of Chinese Academy of Sciences
(XDA2006020102), the NSFC project (41571071 and 21661132003), and Youth Innovation Promotion Association CAS
(2017099). We thank Qomolangma Atmospheric and Environmental Observation and Research Station CAS for providing
radiation data, Dr. Husi Letu and Wenjun Tang for providing Hamawari-8 satellite radiation data. We are also grateful to all
the members who took part in the fieldwork.

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





**Figure and Captions**

**Table 1 The related information about hydro-meteorology observations**

| Parameter | Sensor | accuracy | Location | Duration |
|---|---|---|---|---|
| $T_w$ | HOBO U22-001 | 0.21 $^o$C | Southern basin | 2015.6-2018.5 |
| | | | Northern basin | 2016.6-2017.5 |
| $T_a$<br>RH | HOBO U12-012 | 0.35 $^o$C<br>2.5% | North shoreline | 2015.6-2017.1, 2017.6-2018.5 |
| $R_s$ and $R_a$ | Kipp & Zonen CNR4<br>net radiometer | 5% | Qomolangma<br>Station, CAS | 2015.6-2017.12 |

$T_w$=water temperature; $T_a$=air temperature; RH=relative humidity; $R_s$=solar radiation; $R_a$=downward long wave radiation



**Table 2 Monthly net radiation, total lake heat storage, Bowen ratio and lake evaporation between 2015 and 2017**

| | Net energy (W/m$^2$) | | | Heat storage (W/m$^2$) | | | Bowen Ratio | | | Evaporation (mm/day) | | |
|---|---|---|---|---|---|---|---|---|---|---|---|---|
| | 2015 | 2016 | 2017 | 2015 | 2016 | 2017 | 2015 | 2016 | 2017 | 2015 | 2016 | 2017 |
| May | | 188.5 | 194.8 | | 145.2 | 138.6 | | -0.10 | | | 1.72 | |
| Jun | 217.2 | 214.3 | 224.8 | 157.3 | 191.6 | 181.8 | -0.15 | -0.24 | -0.20 | 2.40 | 0.98 | 1.81 |
| Jul | 198.0 | 185.2 | 218.1 | 123 | 101.0 | 93.4 | -0.02 | 0 | -0.04 | 2.6 | 2.89 | 3.28 |
| Aug | 170.4 | 178.6 | 177.2 | 62.3 | 32.4 | 39.3 | 0.11 | 0.13 | 0.11 | 3.33 | 4.47 | 4.31 |
| Sep | 148.4 | 140.2 | 154.1 | -24.6 | -10.7 | -15.4 | 0.13 | 0.14 | 0.08 | 5.29 | 4.57 | 5.40 |
| Oct | 89.1 | 91.4 | 92.4 | -115 | -87.1 | -86.4 | 0.23 | 0.20 | 0.20 | 5.67 | 5.12 | 5.15 |
| Nov | 34.7 | 34.9 | 34.3 | -140.6 | -193.7 | -199.5 | 0.17 | 0.18 | 0.24 | 5.12 | 6.69 | 6.51 |
| Dec | 17.7 | 16.6 | 19.7 | -192 | -125.3 | -148.5 | 0.26 | 0.14 | 0.20 | 5.78 | 4.22 | 4.88 |






**Table 3 Runoff (m³/s) at the three main rivers at Paiku Co basin in spring and autumn between 2015 and 2017 and their total contribution to lake level increase (mm/day). The measuring dates of runoff are shown in brackets.**

| Rivers | Runoff-2015 | | Runoff-2016 | | Runoff-2017 | |
|---|---|---|---|---|---|---|
| | Spring | Autumn | Spring | Autumn | Spring | Autumn |
| | (6.1~6.2) | (10.6~10.7) | (6.2) | (10.11~10.13) | (5.25~5.28) | (10.14~10.16) |
| Bulaqu | 2.3 | 2.1 | 0.8 | 0.7 | 0.5 | 0.7 |
| Daqu | 0.4 | 2.8 | 1.1 | 1 | 0.5 | 1.2 |
| Barixiongqu | 0.2 | 0.4 | 0.1 | 0.5 | 0.1 | 0.5 |
| Total contribution | 0.89 | 1.64 | 0.62 | 0.71 | 0.62 | 0.74 |

Total contribution is calculated according to the total runoff of the three main rivers and lake area






Figure 1: Monitoring sites for lake level, hydro-meteorology, water temperature, runoff, and total rainfall at Paiku Co basin (A). The upper right (B) denotes the isobath of Paiku Co and the two monitoring sites of water temperature profile. The lower right (C)
denotes the water temperature monitoring at different water depth.







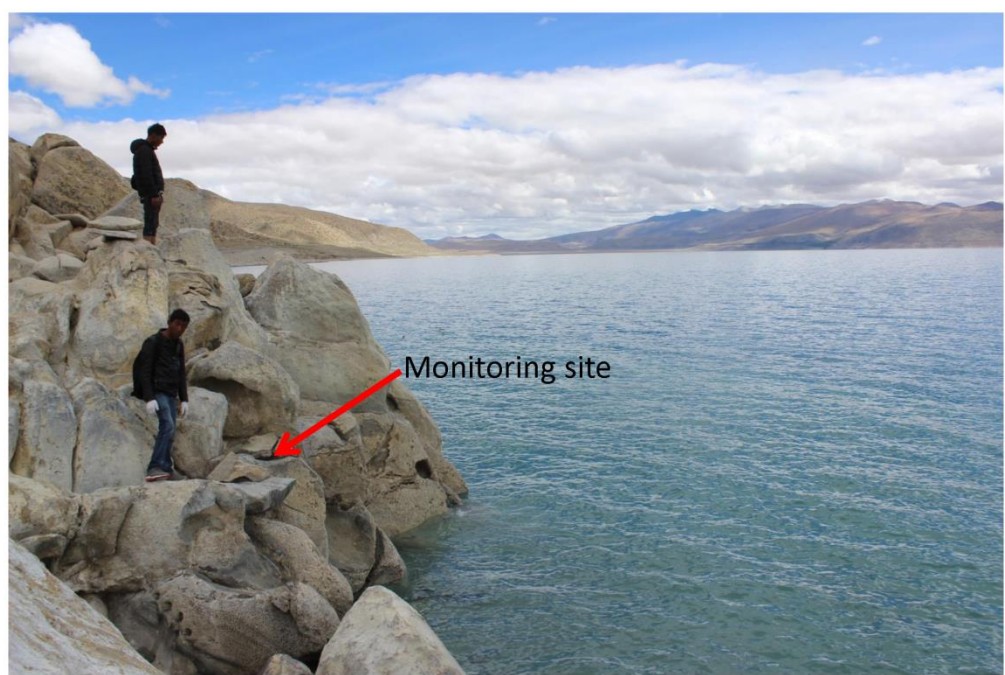

**Figure 2: The monitoring site of air temperature and humidity at the northern shoreline of Paiku Co.**



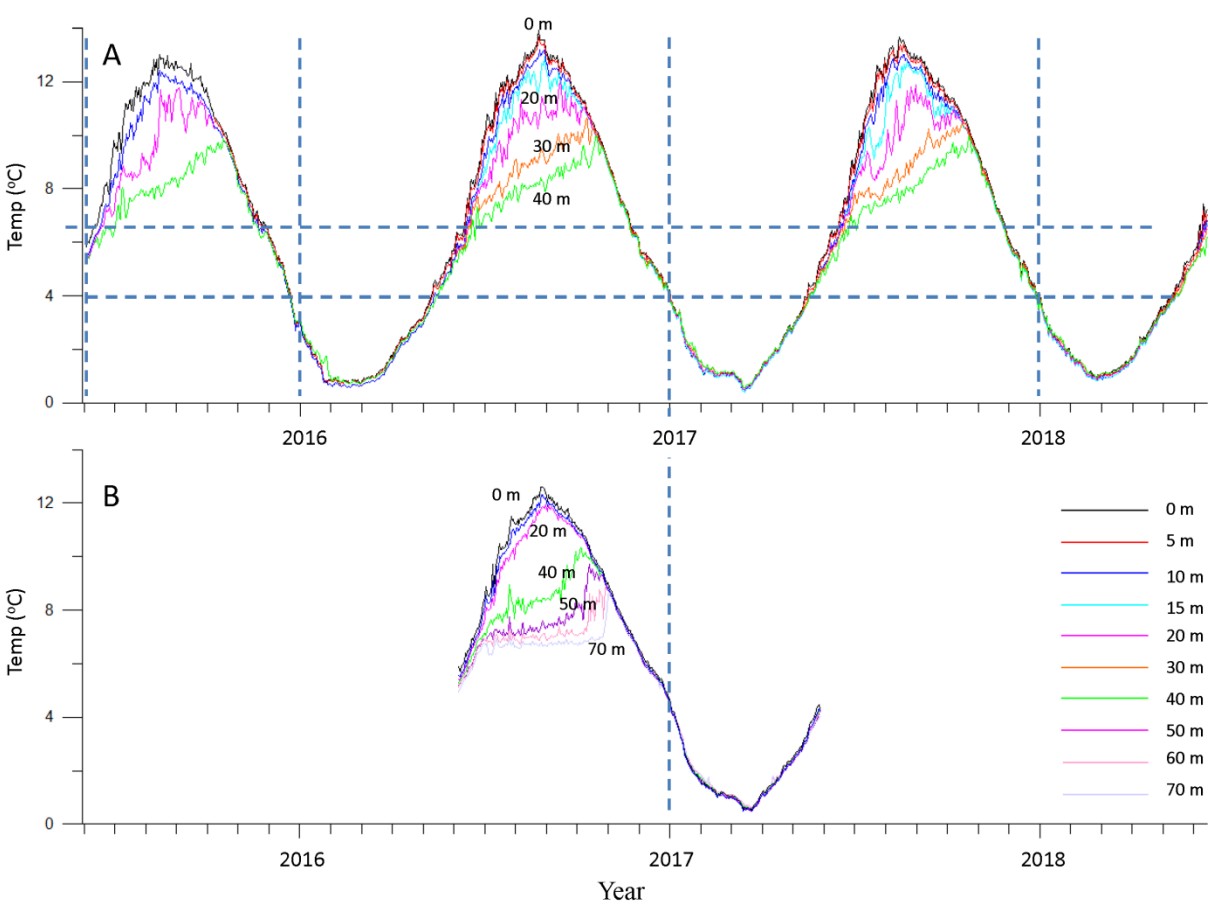

**Figure 3: Time series of daily lake water temperature at different water depths in Paiku Co's southern (upper) and northern (below) basins.**

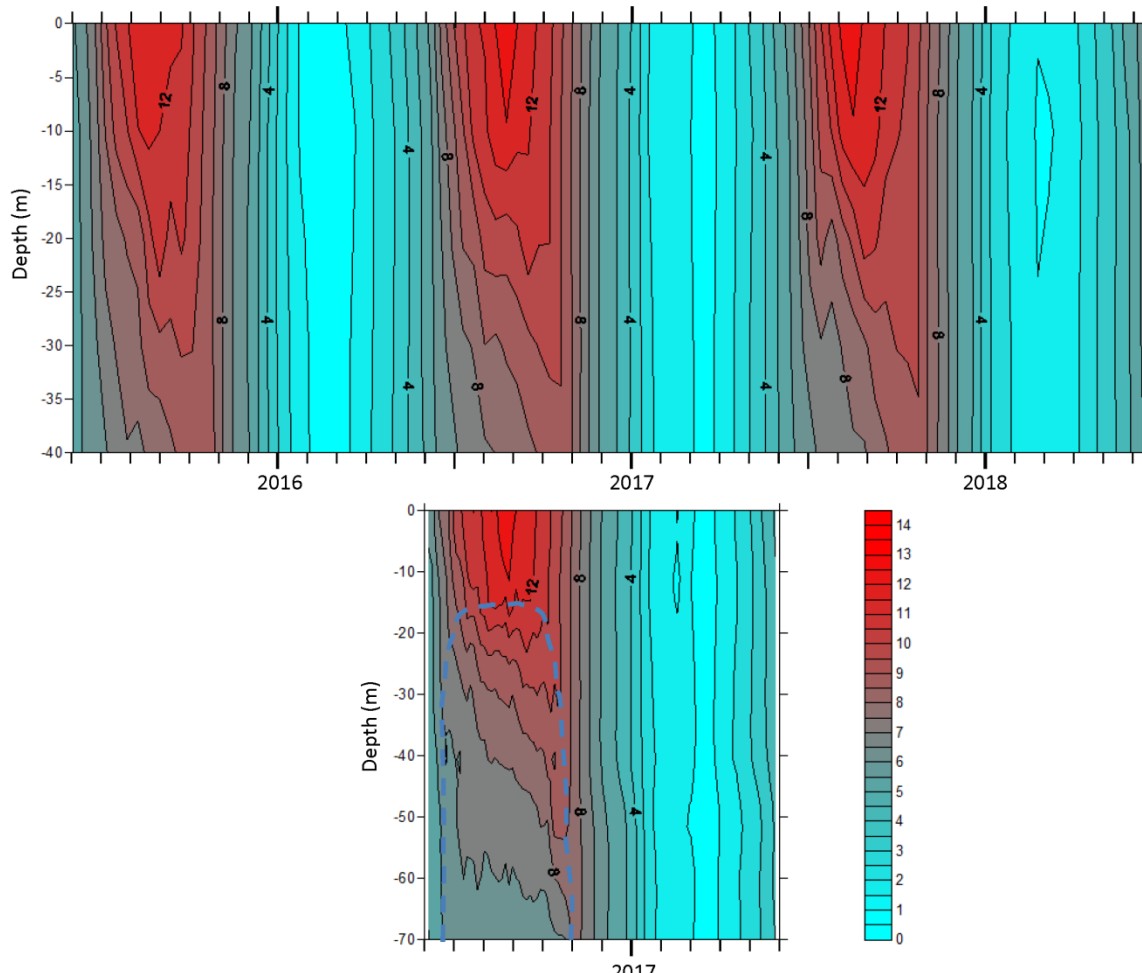

**Figure 4: Depth-time diagram of isotherm (°C) in Paiku Co's southern (upper, 42 m in depth) and northern (below, 72 m in depth) basins between June 2015 and May 2018.**



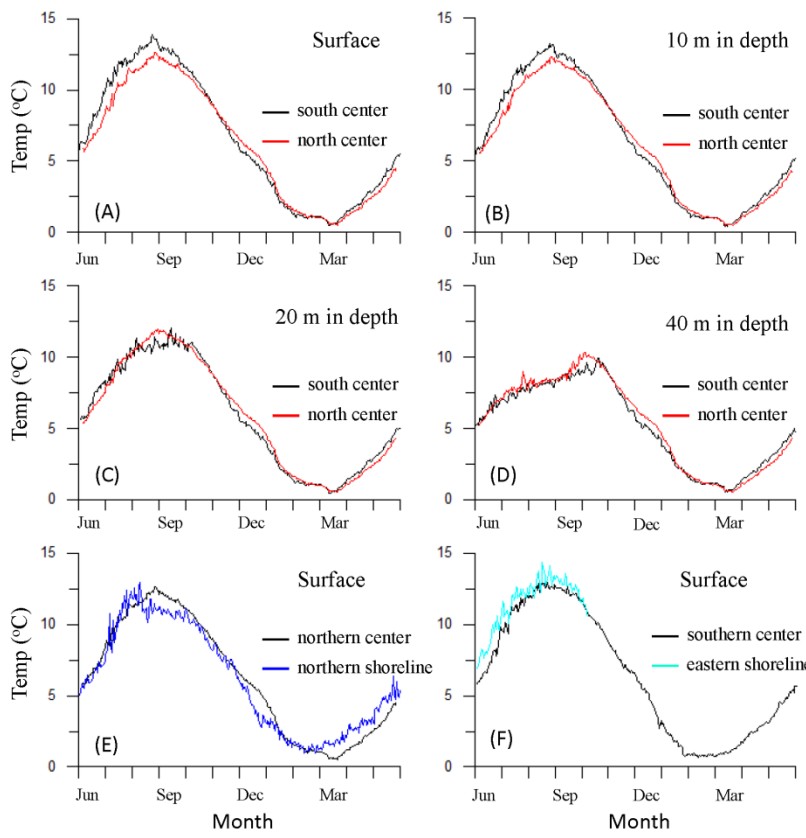


**Figure 5: A comparison of water temperature at the depth of 0 m, 10 m 20 m and 40 m between the southern (black) and northern (red) basins (A-D), and between lake center (black) and shoreline (blue, green E-F).**

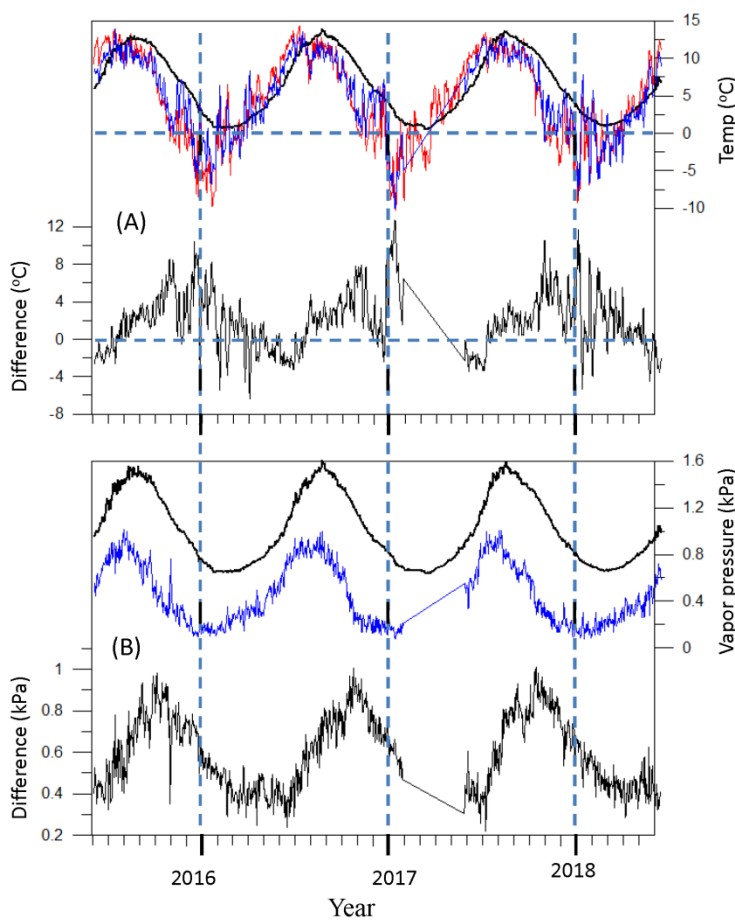

**Figure 6: Time series of hydro-meteorology at Paiku Co. A: Daily lake surface temperature (black), atmosphere temperature at Paiku Co (blue) and Qomolangma station (red), and the difference between lake surface and the overlying atmospheric temperature. B: Actual vapor pressure at lake surface (black) and the overlying atmosphere (blue), and their difference.**



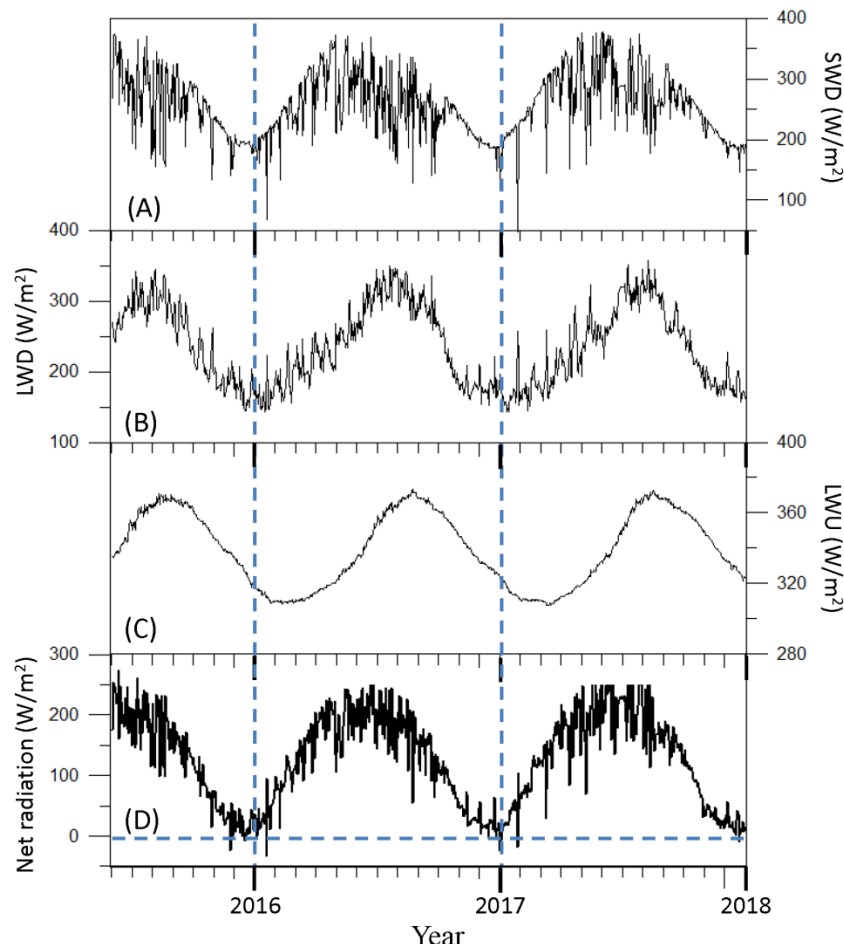

**Figure 7: Time series of daily solar radiation (A), atmospheric long wave radiation to lake (B), long wave radiation**
**emitted from lake (C) and net radiation (D) over Paiku Co.**

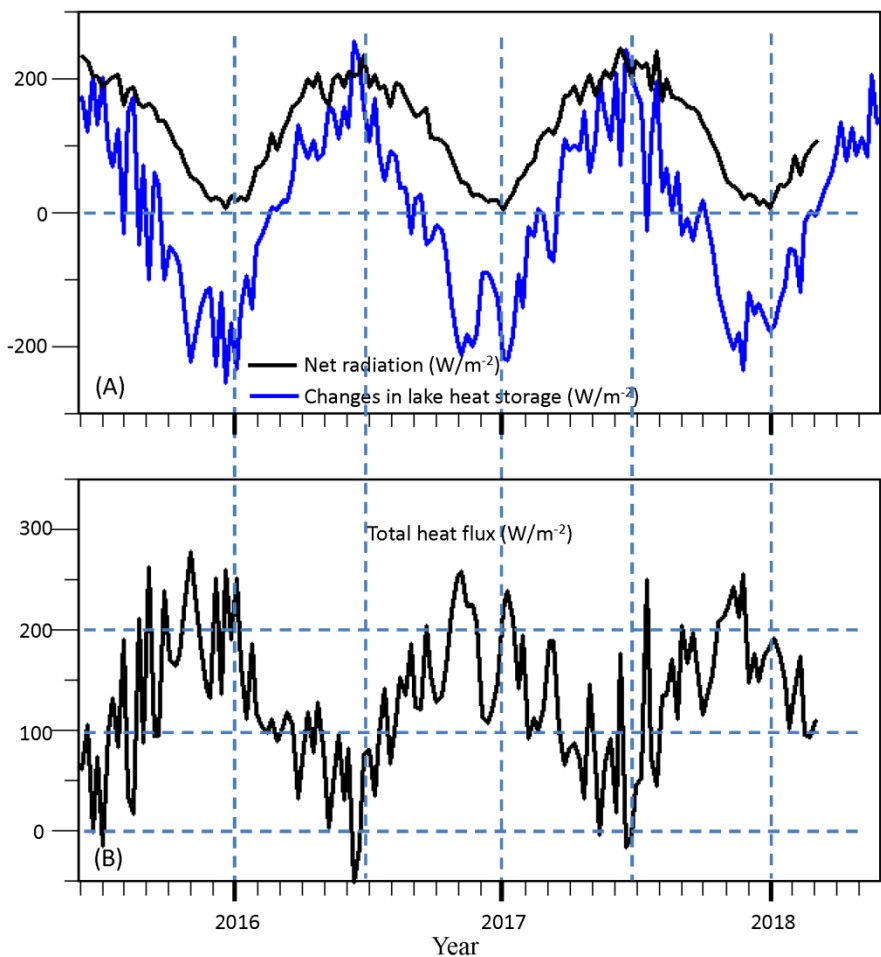

**Figure 8: Time series of weekly net radiation, changes in lake water energy (a), and total heat flux at the lake surface (b).**





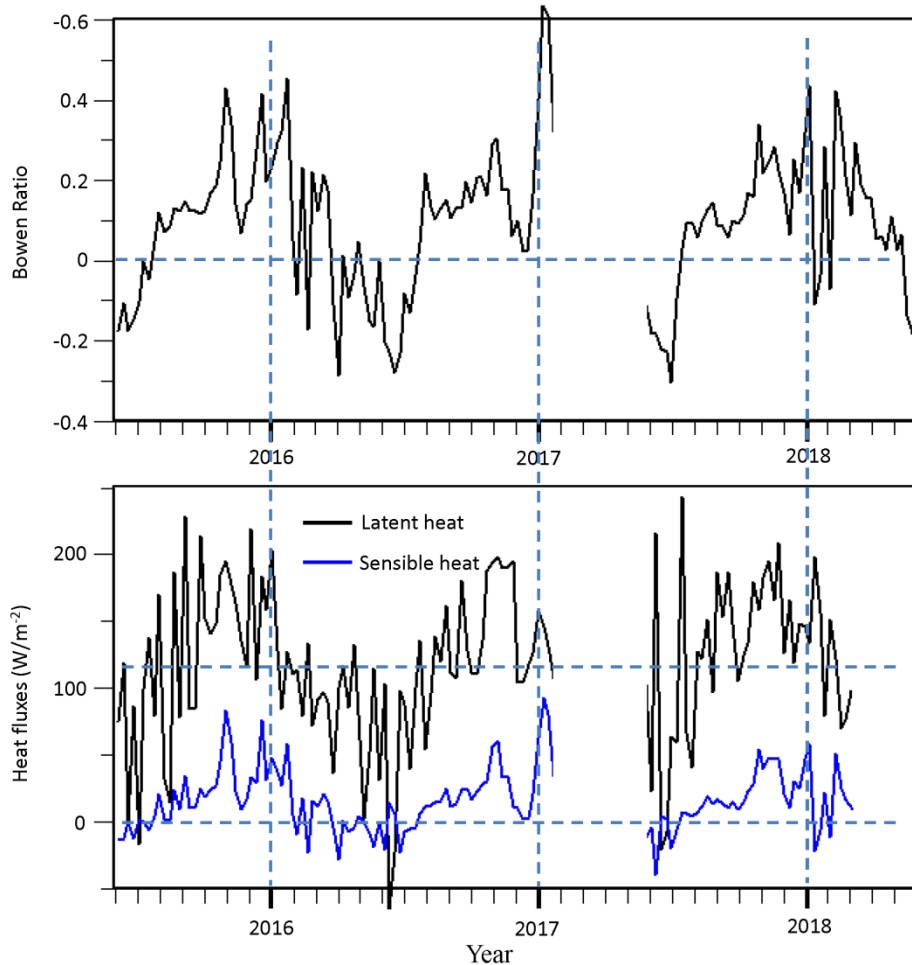

**Figure 9: Time series of Bowen ratio (a), and the latent and sensible heat fluxes at the lake surface (b).**

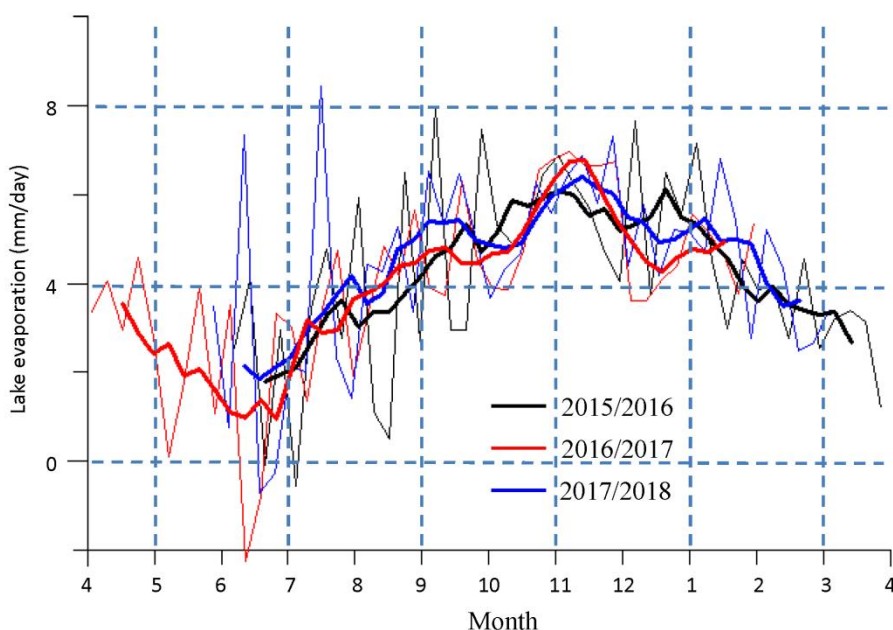

**Figure 10: Weekly lake evaporation (fine lines) at Paiku Co and its 5-point running average (bold lines) between 2015/2016 and 2017/2018.**