# Peer review of "Thermal regime, energy budget and lake evaporation at Paiku Co, a deep alpine lake in the central Himalayas"

_Hydrology and Earth System Sciences, 2019_

## Referee Comment (RC1) · Anonymous Referee #1 · 1 Dec 2019

The manuscript uses in situ hydro-meteorological data to investigate the thermal regime, energy budget components and the evaporation amount of an alpine lake on the Tibetan Plateau. The objectives of this study meets the scope of HESS; the in situ measurements are important for the understanding of these high-elevation lakes and the conclusions are reasonable. I consider the manuscript being an important results for understanding of these high-elevation lakes in this important area. However, there are still some questions needed to be answered and some mistakes needs to be corrected. The following comments are given and a revision is needed for accepting the manuscript to be published. Major comments: 1, Ice cover forms during the winter season in Paiku Co. Thus, what's the influence of ice cover to

your results? The energy budget and evaporation amounts during the ice covered season is quite different from those during ice free season. However, not enough information on ice processes are given in the manuscript. How to consider the energy budget and evaporation amounts during ice covered season in the manuscript? How to get the Bowen ratio during winter season? How much energy may be used for ice processes? As ice surface temperature is not observed, what's the influence of the ice processes to your results. All these information need to be addressed in detail in the revised manuscript. 2, lake level variation is an important content of the manuscript, as shown in Introduction and discussion. However, the lake level measurements are missing in the manuscript. These information should be added in the manuscript. Minor comments: 2, line 24, "significant lake level decrease in post-monsoon season while slight in pre-monsoon". Slight what? 3, line 92, S is the change in lake water energy, but in line 115, it is renamed as lake heat storage, in some place different names are used; it should be kept same. Similarly, what is the "total heat flux" in line 238 and 360? 4, In equation (3), Ra is the longwave radiation from lake surface, $\varepsilon$a is the atmospheric emissivity, it should be water emissivity. However, in equation (2), it is defined as downward longwave radiation to lake. it should be corrected. 5, In line 107, I think it is inappropriate to define Bowen ratio by "Gianniou and Antonopouls, 2007", some classic reference should be given here. 6, In line 118, the definition of $\Delta T$ is not clear? How many layers are defined in vertical direction? 7, Line 135, "the largest temperature difference". Temperature difference between which layers. Similarly, line 137, what is the gradient between which layer? 8, Line 172-175, water circulation along the south-north transection is not evidenced by the observations. Ever give evidence or remove the sentence. 9, Line 187-191, the comparison of the two in situ measurements is not convincing, as the environment and other background information are quite different. Thus, I suggest to remove this part, or give much more information on the comparison. 10, In line 257-258, as change in lake heat storage has quite similar variation with that of net radiation. Why a positive correlation is obtained between lake evaporation and water heat storage change, but

a negative correlation with net radiation. 11, Figure 1 and Figure 2 can be combined together; Figure 3, A and B is given in the figure, but not in the notes; Figure 5, 10m and 20m comma is needed; Figure 8, a and b are used, but in figure it is (A) AND (B);Figure 9, Bowen ratio is given also for winter season, but is may not fit for winter ice covered season. 12, In line 63 "oC"; "W/m2", I think it is better to use "W m-2";

Please also note the supplement to this comment:
https://www.hydrol-earth-syst-sci-discuss.net/hess-2019-421/hess-2019-421-RC1-supplement.pdf

---

## Referee Comment (RC2) · Anonymous Referee #2 · 5 Dec 2019

General comments: This paper revealed the energy balance of a deep lake in Tibetan Plateau, one of the least studied regions on earth. Thus in general, the addition of newly obtained data and their analysis is welcomed and could be scientifically significant. However, I have a concern about the accuracy of energy balance determination in this study. Although the authors discuss uncertainty of lake evaporation estimates, I suspect that uncertainty is much larger than their estimate due to the items which authors did not deal with. Details are given in the following specific comments.

Specific comments:

1. Introduction: The authors should explain why lake level and hydrological processes

in Tibetan Plateau (TP) are important. What kinds of practical and scientific contributions can be made by studying these components over there? Similarly, what are the particular importance of deep lakes TP in comparison with deep lakes in other regions (or other alpine areas)?

2. L76-84. The use of temperature and humidity measured at this location and by this instrument for the purpose of calculating Bowen ratio (Bo) is questionable.

(1) Location

- It is quite possible that this location is outside the internal boundary layer which develops over the lake's surface, particularly when wind direction is from land surface to lake. In order to obtain meaningful Bo values, it is necessary to use measurements within the boundary layer. Note also that Fig.2 should be replaced with a photo showing this location with the actual instrument installed.

- The surface temperature of massive rocks, above which instrument was placed, can be very high during daytime in comparison with air temperature. Thus, the instrument could have been exposed to strong infrared-radiation from rocks. This is a source of measurement errors if instrument does not have a good radiation shield and ventilation (see below).

- Also, given the size of the lake, it is likely that air temperature and humidity near the southern shorelines are different from other parts of the lake.

(2) Instrument

- I have no experience in using a HOBO U12-012 logger, but the manufacturer states that this is designed for indoor use. It seems there is no ventilation of a sensor. Radiation shield (a data logger housing) may not be good enough to prevent effects from direct sunshine in a field condition. These could result in serious measurement errors when it is used outdoors. Authors should explain how (in)accurate their measurements are under their measurement condition and indicate resulting possible errors in flux

estimation.

- The sensor specification states the accuracy of $\pm0.35°C$ for temperature and $\pm2.5\%$ for RH (from10% to 90%). They are not particularly high. The accuracy of the water temperature sensor is $\pm0.2°C$. What would be the resulting accuracy of Bo and fluxes? The final possible error of the estimated fluxes would be due to (1) plus (2).

(3) Independent estimates: if there are wind speed data available, authors may try to apply bulk methods to estimate sensible and latent heat fluxes and compare them with those from the Bowen ratio/energy balance method.

3. L93-94 "For large and deep lake, the components G and AV are small enough to be neglected". This is not automatic, particularly for AV. Whether or not the statement is valid should depend on relative amount of inflow and storage, and respective energy advection and stored energy. For example, when a huge amount of melted snow near zero degrees discharge into a warmer lake late spring to early summer, this can be a substantial energy advection. To clearly indicate that they can be ignored, authors should give supporting evidence for that (e.g., amount of river discharge, river water temperature, etc.).

4. L103-106. Authors assumed Ts=Tw "because surface water can be mixed quickly by wind in the afternoon" and used Tw for their flux estimation. Please show the data to validate this statement. If no data are available, authors may want to add an argument that a small difference between Ts and Tw does not produce large estimation errors of Bo and fluxes. In general, Ts is not equal to Tw even under windy conditions (see., e.g., Prats et al., Earth Syst. Sci Data, 10, 727-743, 2018).

5. Eq.(5) to calculate heat storage change. What is the accuracy of this estimate? Error sources could be (1) measurement error of water level, (2) accuracy of isobath and water volume estimation, in addition to estimation error of mean water temperature of the lake. I assume water density and heat capacity are missing in this equation. $\Delta V$ is the lake volume (and not change), and therefore delta symbol is not necessary. To

make the unit of S in W/m2, I think eq(5) needs to be divided by the lake surface area.

6. L150-158. It is desirable to give comparisons with lakes other than those in TB, to highlight whether or not the thermal structure of lakes in TB is different from those lakes with similar dimensions in other parts of the world.

7. L170-174. "water circulation"; this is an interesting point. Are there any supporting data for the presence of such circulation?

8. L180-183. "large error...if water temperature data collected at the shoreline are used..."; It is also true that some errors can result if only water temperature measured at a central part of the lake is used (and ignore shoreline areas having different temperature) to estimate evaporation of the whole lake.

9. L269 "mean deviation of 1.3 W/m2."; this is a surprisingly small difference. The authors may want to add information on the accuracy of the estimated daily solar radiation by the Himawari-8 satellite data to enhance the credibility of the small difference. By the way, if there are estimates of daily solar radiation at Paiku Co, why not use them for estimating evaporation?

10. L272-L277 "The actual solar radiation at Paiku Co can be considerably overestimated due to the blocking effect of the surrounding mountains around the lake"; I think this type of effects can be estimated by using GIS software such as ArcGIS with DEM as an input.

11. L277-278 "5.4 mm/day.... 3.8 mm/day"; this is a large difference. In fact, the difference (1.6 mm/day) can be translated into 192 mm/(4 months). I suspect that this is closer to actual errors of evaporation estimates than the estimated error given in L295.

12. Chapter 4.2, Again, it is desirable to give comparisons with lakes other than those in TB, to highlight whether or not the thermal structure of lakes in TB is different from those lakes with similar dimensions in other parts of the world.

13. L333-335. "In-situ observations of runoff at the three main rivers indicate that the surface runoff had weak impact on lake level changes.....(Table 3)"; Discharge values in Table 3 are only for short durations. Are those periods during baseflow? What would happen in case of rainfall-runoff events, or snow melting discharge?

14. L373 "Bird B.W. polished the language." I am not familiar with the author's guideline for HESS, but personally, I do not think this is a good reason to make Bird B.W be a co-author.

15. Fig.9; Why there are a large fluctuation when calculation was made with weekly averaged data? Do peaks correspond with week-long sunny periods between rainy events or cloudy conditions?

Technical corrections:

L62: Correct degree sign.

L95: Eq. (2); this equation is confusing. Ra is stated as "downward longwave radiation" in L 97 while Eq. (3) specifies Ra as upward longwave radiation from lake. Downward longwave radiation should be Ra, part of which $(0.03Ra=(1-\varepsilon)Ra)$ is reflected by the surface and the surface also emits upward longwave radiation ($\varepsilon\sigma Ts^4$). So the final equation of longwave radiation balance should be $Ra-(1-\varepsilon)Ra+\varepsilon\sigma Ts^4=\varepsilon Ra+\varepsilon\sigma Ts^4$.

L96: "Rsr.... which is taken as 0.07"; this does not make sense. Could it be 0.07Rs?

L97-98 "Rar...., which is taken as 0.03"; this does not make sense. Could it be 0.03Ra?

L103 "atmospheric emissivity"; the word "atmospheric" is not needed if it indicates water surface emissivity.

Fig.1; Is a black polygon surrounding Paiku Co a watershed divide? Add this information in legend. Show more clearly where incoming rivers are. Also, add information on the elevation of surrounding areas.

Fig.9: Explain what dashed blue lines indicate, please.

[Figure]

---

## Author Comment (AC1) · 3 Jan 2020

**Reviewer #1**

The manuscript uses in situ hydro-meteorological data to investigate the thermal regime, energy budget components and the evaporation amount of an alpine lake on the Tibetan Plateau. The objectives of this study meets the scope of HESS; the in situ measurements are important for the understanding of these high-elevation lakes and the conclusions are reasonable. I consider the manuscript being an important results for understanding of these high-elevation lakes in this important area. However, there are still some questions needed to be answered and some mistakes needs to be corrected. The following comments are given and a revision is needed for accepting the manuscript to be published.

Reply: Thanks for the comments. These comments are very helpful to improve the manuscript. We will revise the manuscript carefully according to these comments.

Major comments:

1, Ice cover forms during the winter season in Paiku Co. Thus, what's the influence of ice cover to your results? The energy budget and evaporation amounts during the ice covered season is quite different from those during ice free season. However, not enough information on ice processes are given in the manuscript. How to consider the energy budget and evaporation amounts during ice covered season in the manuscript? How to get the Bowen ratio during winter season? How much energy may be used for ice processes? As ice surface temperature is not observed, what's the influence of the ice processes to your results. All these information need to be addressed in detail in the revised manuscript.

Reply: We will add section 3.7 in revision to discuss the impact of lake ice phenology on lake evaporation and lake level change. In winter 2013/2014, Paiku Co was fully frozen up between middle January and middle April, as indicated by Landsat satellite images. During this period, lake evaporation is very low because the lake ice can effectively prohibit lake evaporation. In the winter 2014/2015, only the southern part of Paiku Co was frozen up. After that, only a small part around the shoreline was frozen, and the lake center was not frozen up any more. During our study period (June 2015-May 2018), lake ice occurs only at the shoreline intermittently. So the impact of lake ice on lake evaporation is not considered in this study.

The impact of lake evaporation on lake level changes will be discussed in section 3.7 in the revision. When the lake surface was frozen in winter 2013/2014, lake level at Paiku Co was very stable. However, when there was no or only small part of lake ice at Paiku Co since the winter 2015/2016, lake level dramatically decreased by 132 mm on average between middle January and middle April. This indicates that lake evaporation increased significantly when there is no lake ice. Therefore, change in lake ice phenology may have significant impact on lake water balance. As has been addressed by Lei et al (2018), Paiku Co has been shrinking since the 1970s. The disappearance of lake ice under climate warming may probably lead to more negative lake water balance and more rapid lake shrinkage in the future.

2, lake level variation is an important content of the manuscript, as shown in Introduction and discussion. However, the lake level measurements are missing in the manuscript. These information should be added in the manuscript.

Reply: Lake level changes at Paiku Co between 2013 and 2018 will be added in Fig. 11 in the

revision. The difference of lake level changes with or without lake ice will be discussed in section 3.7 and 4.3. Lake level at Paiku Co was very stable during the period when the lake surface was frozen (e.g. the winter of 2013/2014). However, lake level dropped significantly by 13.2 cm on average in winter when the lake surface was not completely frozen. This change in lake ice may have significant impact on the long-term lake water balance.

Minor comments:
2, line 24, "significant lake level decrease in post-monsoon season while slight in pre-monsoon". Slight what?
Reply: We will address the different rate of lake level changes in the revision.

3, line 92, S is the change in lake water energy, but in line 115, it is renamed as lake heat storage, in some place different names are used; it should be kept same. Similarly, what is the "total heat flux" in line 238 and 360?
Reply: Thanks for pointing out this. Lake heat storage is used in the revision. Total heat flux is the sum of sensible and latent heat fluxes.

4, In equation (3), Ra is the longwave radiation from lake surface, "a is the atmospheric emissivity, it should be water emissivity. However, in equation (2), it is defined as downward longwave radiation to lake. It should be corrected.
Reply: Thanks for pointing out this. We have corrected it in the revision.

5, In line 107, I think it is inappropriate to define Bowen ratio by "Gianniou and Antonopouls, 2007", some classic reference should be given here.
Reply: We will add some classic references about Bowen ration.

6, In line 118, the definition of _T is not clear? How many layers are defined in vertical direction?
Reply: We have addressed this in more detailed in the revision. Changes in lake heat storage are calculated at an interval of 5 m and therefore there are 13 layers in vertical direction. Lake volume is acquired according to the 5 m isobaths. Lake water temperature at each layer is taken as the average value at the top and bottom layer.

7, Line 135, "the largest temperature difference". Temperature difference between which layers. Similarly, line 137, what is the gradient between which layer?
Reply: Usually, lake water temperature is stable in surface layer and bottom layer, but it changes greatly at thermocline. The temperature difference is the difference of thermocline.

8, Line 172-175, water circulation along the south-north transection is not evidenced by the observations. Ever give evidence or remove the sentence.
Reply: We agree that further evidence is needed to confirm the water circulation. We will discuss it in the revision.

9, Line 187-191, the comparison of the two in situ measurements is not convincing, as the

environment and other background information are quite different. Thus, I suggest to remove this part, or give much more information on the comparison.

Reply: We agree that further evidence is needed to confirm this. We will remove it in the revision.

10, In line 257-258, as change in lake heat storage has quite similar variation with that of net radiation. Why a positive correlation is obtained between lake evaporation and water heat storage change, but a negative correlation with net radiation.

Reply: Lake heat storage exhibits similar seasonal variations with that of net radiation. When the net radiation is high between May and July, a lot of the energy is used to heat the lake water. Lake evaporation during this period is also low because only a small portion of energy is used as latent heat. When the net radiation is low between November and December, the lake water releases a lot of energy to the overlying atmosphere. Lake evaporation during this period is high because only a lot of energy from lake is also used as latent heat.

11, Figure 1 and Figure 2 can be combined together; Figure 3, A and B is given in the figure, but not in the notes; Figure 5, 10m and 20m comma is needed; Figure 8, a and b are used, but in figure it is (A) AND (B);Figure 9, Bowen ratio is given also for winter season, but is may not fit for winter ice covered season. 12, In line 63 "oC"; "W/m2", I think it is better to use "W m-2";

Reply: Thanks for the suggestions. We will revise these in the revision.

---

## Author Comment (AC2) · 3 Jan 2020

**Reviewer #2**

General comments: This paper revealed the energy balance of a deep lake in Tibetan Plateau, one of the least studied regions on earth. Thus in general, the addition of newly obtained data and their analysis is welcomed and could be scientifically significant. However, I have a concern about the accuracy of energy balance determination in this study. Although the authors discuss uncertainty of lake evaporation estimates, I suspect that uncertainty is much larger than their estimate due to the items which authors did not deal with. Details are given in the following specific comments.

Reply: Thanks for the constructive comments. They are very helpful to improve the manuscript. We will revise the manuscript carefully according to these comments. In fact, your concerns are also our concerns. Further works are expected to set up an observing platform in the lake to improve the understanding of energy and water budget.

Specific comments:

1, Introduction: The authors should explain why lake level and hydrological processes in Tibetan Plateau (TP) are important. What kinds of practical and scientific contributions can be made by studying these components over there? Similarly, what are the particular importance of deep lakes TP in comparison with deep lakes in other regions (or other alpine areas)?

Reply: We will address these issues in the revision in more detailed. About the importance of this study, we will add one paragraph in the introduction. 'The Tibetan Plateau (TP) hosts the greatest concentration of high-altitude inland lakes in the world. More than 1200 lakes (>1 km$^2$) are distributed on the TP, with a total lake area of ~45000 km$^2$. Since the late 1990s, most lakes on the interior TP expanded dramatically, with increase in total lake area by >20% between the 1990s and 2010s. In contrast, lakes on the southern TP shrank dramatically during the past decades. These lake expansion or shrinkage indicates significant changes in the regional water cycle occurred on the TP in response to recent climate changes and cryosphere melting. Investigation of lake water budget at specific lakes is needed to understand changes in hydrological processes on the TP under climate warming.'

About the scientific contribution of this study (section 4.3): 'This study is important to understand the different lake level seasonality in high elevation region. Previous studies show that there were different amplitudes of lake level fluctuations even in similar climate regimes. For example, lake level at Nam Co and Zhari Namco, two large and deep lakes on the central TP, decreased considerably by 0.3-0.5 m in cold season (October to December), while lake level at two nearby small lakes, Bam Co and Dawa Co, decreased slightly by of 0.1-0.2 m during the same period. For deep lakes (e.g. Paiku Co, Nam Co and Zhari Namco), the latent heat flux (lake evaporation) over lake surface may lag the solar radiation by several months due to the large heat capacity of lake water. For this kind of lake, the lake level drop is most dramatic in the autumn and early winter when lake evaporation is high. For shallow lakes, the latent heat flux closely follows solar radiation, with high lake evaporation during pre-monsoon and monsoon seasons, and low lake evaporation during post monsoon season (Morrill et al., 2004). Meanwhile, the shallow lakes freeze up 1-2 months earlier than deep

lakes. When the lake surface is covered by ice, lake evaporation is effectively prohibited. Consequently, lake level decreased at a much slower rate in shallow lakes compared with deep lakes.'

2. L76-84. The use of temperature and humidity measured at this location and by this instrument for the purpose of calculating Bowen ratio (Bo) is questionable.
(1) Location
- It is quite possible that this location is outside the internal boundary layer which develops over the lake's surface, particularly when wind direction is from land surface to lake. In order to obtain meaningful Bo values, it is necessary to use measurements within the boundary layer. Note also that Fig.2 should be replaced with a photo showing this location with the actual instrument installed.
- The surface temperature of massive rocks, above which instrument was placed, can be very high during daytime in comparison with air temperature. Thus, the instrument could have been exposed to strong infrared-radiation from rocks. This is a source of measurement errors if instrument does not have a good radiation shield and ventilation (see below).
- Also, given the size of the lake, it is likely that air temperature and humidity near the southern shorelines are different from other parts of the lake.
Reply: We agree that instrument should be installed in a right place. We will address the location of the instrument in more detailed in the revision. Fig.2 will be replaced by the Figure below, which shows more detailed information about the installation of the instrument. Paiku Co is a deep lake and has steep shoreline. It is difficult to install the instrument in the lake center. The logger was installed in an outcrop ~2 m above the lake surface at the north part of the lake. The instrument is just under a rock where there is a hole facing the lake. We believe that this is an ideal place to install the instrument. The meteorological condition over the lake surface can be well recorded.

[Figure]

Figure 1. Location of the HOBO logger at the northern shoreline of Paiku Co.

(2) Instrument
- I have no experience in using a HOBO U12-012 logger, but the manufacturer states that this is designed for indoor use. It seems there is no ventilation of a sensor. Radiation shield (a data logger housing) may not be good enough to prevent effects from direct sunshine in a field condition. These could result in serious measurement errors when it is used outdoors. Authors should explain how (in) accurate their measurements are under their measurement condition and indicate resulting possible errors in flux estimation.
- The sensor specification states the accuracy of _0.35_C for temperature and _2.5% for RH

(from10% to 90%). They are not particularly high. The accuracy of the water temperature sensor is _0.2_C. What would be the resulting accuracy of Bo and fluxes? The final possible error of the estimated fluxes would be due to (1) plus (2).

Reply: It is true that the instrument we used in this study is designed for indoor use. We selected this instrument for measuring air temperature and humidity because it is cheap and easy to install. In fact, the instrument is installed just under a big stole where there is good ventilation, so the meteorological condition over the lake surface can be well recorded. The instrument has an accuracy of 0.35 $^{\circ}$C for air temperature and 2.5% of relative humidity. We will evaluate the accuracy of Bowen ratio caused by this error.

(3) Independent estimates: if there are wind speed data available, authors may try to apply bulk methods to estimate sensible and latent heat fluxes and compare them with those from the Bowen ratio/energy balance method.

Reply: From the beginning of this study, energy budget is designed to calculate lake evaporation. Wind speed in the study area is not available. A lot of experience is also needed if bulk method is also used to estimate sensible and latent heat fluxes. We will validate lake evaporation independently by comparing with lake level changes and runoff.

3. L93-94 "For large and deep lake, the components G and AV are small enough to be neglected". This is not automatic, particularly for AV. Whether or not the statement is valid should depend on relative amount of inflow and storage, and respective energy advection and stored energy. For example, when a huge amount of melted snow near zero degrees discharge into a warmer lake late spring to early summer, this can be a substantial energy advection. To clearly indicate that they can be ignored, authors should give supporting evidence for that (e.g., amount of river discharge, river water temperature, etc.).

Reply: Paiku Co is a large and deep lake. Runoff only accounts for less than 2% of lake water storage. Water temperature and level are also recorded at the three largest rivers in Paiku Co basin (Figure 1). We can compare the water temperatures at rivers with lake water temperature. The heat input from rivers can be roughly evaluated. We found that the influence of Av on the lake heat storage is small and can be neglected. We will address this in section 2.2 in more detailed in the revision.

4. L103-106. Authors assumed Ts=Tw "because surface water can be mixed quickly by wind in the afternoon" and used Tw for their flux estimation. Please show the data to validate this statement. If no data are available, authors may want to add an argument that a small difference between Ts and Tw does not produce large estimation errors of Bo and fluxes. In general, Ts is not equal to Tw even under windy conditions (see., e.g., Prats et al., Earth Syst. Sci Data, 10, 727-743, 2018).

Reply: We agree that Ts is not equal to Tw. In this study, we do not measure the surface water temperature and lake water temperature at 0.4~0.8 m is used to represent the surface water temperature. However, there is small difference between them and this difference does not produce large estimation error of Bo and heat fluxes.

5. Eq.(5) to calculate heat storage change. What is the accuracy of this estimate? Error

sources could be (1) measurement error of water level, (2) accuracy of isobaths and water volume estimation, in addition to estimation error of mean water temperature of the lake. I assume water density and heat capacity are missing in this equation. _V is the lake volume (and not change), and therefore delta symbol is not necessary. To make the unit of S in W/m2, I think eq(5) needs to be divided by the lake surface area.

Reply: Thanks for pointing out this. The equation should be 'S $= \frac{\sum_{i=0}^{72.8} c_w \times \rho_w \times \Delta V_i \times \Delta T_i}{A_l}$'. Here $c_w$ is the specific heat of water (J kg$^{-1}$ K$^{-1}$), $\rho_w$ is water density (kg m$^{-3}$), $\Delta V_i$ is the lake volume at certain depth, and $\Delta T_i$ is water temperature change at the same depth, $A_l$ is lake area (m$^2$). Changes in lake heat storage are calculated at an interval of 5 m and therefore there are 13 layers in vertical direction. Lake volume is acquired according to the 5 m isobaths. Lake water temperature at each layer is taken as the average value between the top and bottom lay. We believe that there is no need to estimate the accuracy of lake heat storage because both the lake bathymetry and lake water temperature are all in-situ measurement.

6. L150-158. It is desirable to give comparisons with lakes other than those in TB, to highlight whether or not the thermal structure of lakes in TB is different from those lakes with similar dimensions in other parts of the world.
Reply: We will compare the thermal structure at Paiku Co with other lakes, not only from the Tibetan Plateau, but also from the other parts of the world, for example, the Great Slave Lake in Canada (Schertzer et al., 2003) and Issyk-Kul Lake in central Asia.

7. L170-174. "water circulation"; this is an interesting point. Are there any supporting data for the presence of such circulation?
Reply: We will discuss this in more detailed in the revision. 'This contrasting pattern of water temperature between the surface and bottom layers occurs in early autumn when the bottom water temperature reaches to its highest. This indicates that deeper water convection occurs in the northern basin than the southern basin when the water temperature gradient on vertical profile also starts to decrease during this period.' However, more evidence is needed to confirm this.

8. L180-183. "large error...if water temperature data collected at the shoreline are used..."; It is also true that some errors can result if only water temperature measured at a central part of the lake is used (and ignore shoreline areas having different temperature) to estimate evaporation of the whole lake.
Reply: We agree with this that both have their representativeness. But Paiku Co is a deep lake with an average water depth of 41 m. Lake water temperature at the center of lake can more represent the average state of lake water temperature.

9. L269 "mean deviation of 1.3 W/m2."; this is a surprisingly small difference. The authors may want to add information on the accuracy of the estimated daily solar radiation by the Himawari-8 satellite data to enhance the credibility of the small difference. By the way, if there are estimates of daily solar radiation at Paiku Co, why not use them for estimating evaporation?

Reply: In this study, we do not have in-situ observations of solar radiation at Paiku Co, so we have to use the solar radiation and long-wave atmospheric radiation at Qomolangma Station, which is about 150 km away from Paiku Co. Both sites are in dry environment.Solar radiation at the two sites exhibited very similar seasonal fluctuations ($R^2$=0.55, P<0.001) with standard deviation of 23.9 W/m$^2$ and mean deviation of 1.3 W/m$^2$. In order to reduce the error caused by regional difference, weekly averaged radiation was used to calculate lake evaporation. The satellite short wave radiation derived from Himawari-8 satellite is not used for the lake evaporation estimation in this study because this dataset can not provide downward longwave radiation.

[Figure]

 A comparison of solar radiation at Paiku Co and Qomolangma Station derived from Hamawari-8

10. L272-L277 "The actual solar radiation at Paiku Co can be considerably overestimated due to the blocking effect of the surrounding mountains around the lake"; I think this type of effects can be estimated by using GIS software such as ArcGIS with DEM as an input.
Reply: As a large lake, we do not consider the blocking effect of the surrounding mountains in this study because it mainly occurs in shoreline.

11. L277-278 "5.4 mm/day.... 3.8 mm/day"; this is a large difference. In fact, the difference (1.6 mm/day) can be translated into 192 mm/(4 months). I suspect that this is closer to actual errors of evaporation estimates than the estimated error given in L295.
Reply: We will evaluate the difference between lake level changes and lake evaporation in more detailed in revision (section 4.1). The runoff measurement at the three large rivers (Fig. 1) makes it possible to compare the lake evaporation with lake level decrease. In pre-monsoon season (mid-April to mid-May), lake evaporation (1.7 mm/day) was quite similar with the decreasing rate of lake level (1.8 mm/day). The high consistency between lake evaporation and lake level decrease confirms the reliability of lake evaporation estimation. In post-monsoon season (October to January), lake evaporation (5.4 mm/day) is considerably higher than the rate of lake level decrease (3.8 mm/day). This discrepancy may be due to the contribution of precipitation and surface runoff (Tab. 3). As shown in Tab. 3, runoff at the three large rivers can contribute to lake level increase by 0.7~1.6 mm/day in October. The

impact of lake evaporation on lake level changes can be partially offset. According to this difference of 0.9 mm/day in post-monsoon, the error of lake evaporation is estimated to be 82.8 mm/year.

12. Chapter 4.2, Again, it is desirable to give comparisons with lakes other than those in TB, to highlight whether or not the thermal structure of lakes in TB is different from those lakes with similar dimensions in other parts of the world.
Reply: We will compare the thermal structure at Paiku Co with other lakes in other parts of the world, e.g. the Great Slave Lake (Schertzer et al., 2003) and Issyk-Kul Lake.

13. L333-335. "In-situ observations of runoff at the three main rivers indicate that the surface runoff had weak impact on lake level changes.....(Table 3)"; Discharge values in Table 3 are only for short durations. Are those periods during baseflow? What would happen in case of rainfall-runoff events, or snow melting discharge?
Reply: The runoff measurement was mainly conducted in late May or early October when the water level is still low. Besides discharge measurement in the three rivers, water level is also records by using HOBO water level loggers. We found that this discharge can approximately represent the average state in spring and autumn. It can still not represent the baseflow because there is almost no surface runoff between January and March.

14. L373 "Bird B.W. polished the language." I am not familiar with the author's guideline for HESS, but personally, I do not think this is a good reason to make Bird B.W be a co-author.
Reply: We believe that Bird BW should be included because he also contributed a lot to data analysis, besides language polishing.

15. Fig.9; Why there are a large fluctuation when calculation was made with weekly averaged data? Do peaks correspond with week-long sunny periods between rainy events or cloudy conditions?
Reply: The large fluctuation may indicate a different climate condition, such as air temperature, humidity, solar radiation and changes in lake heat storage. The 5 point running average exhibits less fluctuation (Fig. 10). We notice that lake evaporation derived from eddy co-variance method also shows large fluctuation.

Technical corrections:
L62: Correct degree sign.
Reply: Thanks for pointing out this. We have corrected it.

L95: Eq. (2); this equation is confusing. Ra is stated as "downward longwave radiation" in L 97 while Eq. (3) specifies Ra as upward longwave radiation from lake. Downward longwave radiation should be Ra, part of which (0.03Ra=(1-")Ra) is reflected by the surface and the surface also emits upward longwave radiation ("_Ts^4). So the final equation of longwave radiation balance should be Ra-(1-")Ra+"_Ts^4="Ra+"_Ts^4. L96: "Rsr.... which is taken as 0.07"; this does not make sense. Could it be 0.07Rs?
Reply: Thanks for pointing out this. We have changed Ra to Rw in equation (3) and address

this sentence more precisely. '$R_s$ is downward shortwave radiation, $R_{sr}$ is the reflection of solar radiation from lake surface, which is taken as 0.07 $R_s$ in this study (Gianniou and Antonopouls, 2007), $R_a$ is downward longwave radiation to lake, $R_{ar}$ is the reflected longwave radiation from the lake surface, which is taken as 0.03 $R_a$, and $R_w$ is the longwave radiation from the lake surface. The units of the items in Eq (2) are W/m$^2$'.

L97-98 "Rar...., which is taken as 0.03"; this does not make sense. Could it be 0.03Ra?
Reply: Thanks for pointing out this. $R_{ar}$ is the reflected longwave radiation from the lake surface, which is taken as 0.03 $R_a$.

L103 "atmospheric emissivity"; the word "atmospheric" is not needed if it indicates water surface emissivity.
Reply: Thanks for pointing out this. It should be 'water emissivity'.

Fig.1; Is a black polygon surrounding Paiku Co a watershed divide? Add this information in legend. Show more clearly where incoming rivers are. Also, add information on the elevation of surrounding areas.
Reply: We will add locates of the three main rivers and the elevation information in Fig.1 in revision. The information about the catchment boundary will be addressed in the legend.

Fig.9: Explain what dashed blue lines indicate, please.
Reply: The dashed blue lines do not have scientific meaning. They are mainly used for the convenience of readers. We will delete some in the revision.